

# Development and validation of the Exteroceptive Body Awareness (EBA-q) questionnaire

Alisha Vabba[1,2], Giuseppina Porciello[2,3], Maria Serena Panasiti[2,3] and Salvatore Maria Aglioti[1,2]

[1] Department of Psychology, Sapienza University of Rome and CLN2S@Sapienza, Rome, Italy
[2] IRCCS, Santa Lucia Foundation, Rome, Italy
[3] Department of Psychology, "Sapienza" University of Rome, Rome, Italy

## ABSTRACT

The conscious processing of body signals influences higher-order psychological and cognitive functions, including self-awareness. Dysfunctions in the processing of these signals has been connected to neurological and psychiatric disorders characterized by altered states of self-consciousness. Studies indicate that perceiving the body through interoceptive signals (*e.g.*, from internal organs such as heartbeat and breathing) is distinct from perceiving the body through exteroceptive signals (*e.g.*, by relying on visual, tactile and olfactory cues). While questionnaires are available for assessing interoception, there are no validated self-report instruments for measuring bodily exterception. To fill this gap, we performed three studies to develop and validate a novel scale designed to assess bodily self-consciousness based on the processing of exteroceptive bodily signals. Exploratory factor analysis (Study 1, $N = 302$) led to an 18-item questionnaire comprised of four factors. We called this instrument Exteroceptive Body Awareness questionnaire (EBA-q). Confirmatory factor analysis (Study 2, $N = 184$) run on a second sample showed an acceptable fit for a bifactor model, suggesting researchers may use the questionnaire as a unidimensional scale reflecting exteroceptive bodily self-consciousness, or use each of its four sub-scales, reflecting "visuo-tactile body awareness", "spatial coordination", "awareness of body changes" and "awareness of clothing fit". Overall EBA-q showed good internal consistency. Convergent and divergent validity were assessed *via* cross-validation with existing body awareness questionnaires (Study 3, $N = 366$) and behavioral measures (Study 3, $N = 64$) of exteroceptive and interoceptive bodily self-consciousness. Research applications are discussed within a multi-faceted model of exteroception and interoception as distinct, but at the same time interconnected, dimensions of bodily self-consciousness.

## INTRODUCTION

The past few decades have shown an increased scientific interest in the role of the body and bodily signals in understanding human cognition and behavior. This led to the development of the embodied cognition theories, according to which even higher-order cognitive and

Corresponding author
Alisha Vabba,
alisha.vabba@uniroma1.it

emotional processes are grounded in the bodily self (*Goldman & De Vignemont, 2009*). The awareness of owning a body and being the initiators and the controllers of our own actions is known as corporeal awareness (*Berlucchi & Aglioti, 2010*) or bodily self-consciousness (BSC), and relies on the ability of the brain to continuously integrate information about the body originating from different sensory modalities (*Blanke, 2012*). Specifically, seminal theoretical models such as the one proposed by *Freeman & Sherrington (1907)* suggest that BSC is constituted of different sensations, namely the awareness of the body based on signals arising outside the body such as tactile, visual, and auditory, known as exteroception, the awareness the physiological state of the body, including signals from inside the body such as heartbeat, breathing, and gastric function, known as interoception, and the awareness of body position and movements based on the processing of signals from the joints and muscles, known as proprioception (see *Ceunen, Vlaeyen & Diest, 2016*).

These incoming signals are continuously integrated with our prior knowledge and expectations to determine and update our conscious experience of selfhood and even our own identity (*Aspell, Lenggenhager & Blanke, 2011*; *Clark, 2013*; *Ehrsson, 2011*; *Park & Blanke, 2019*; *Porciello et al., 2018*; *Seth, 2013*; *Seth & Friston, 2016*). Indeed, dysfunctions in the processing of body signals have been connected to neurological and psychiatric disorders characterized by altered states of self-consciousness, such as depression, eating disorders, depersonalization, and derealization disorders (*Khalsa et al., 2018*).

A series of behavioral and self-reported measures have been devised to explore individual differences in the capacity to detect bodily signals, as well as the tendency to focus on and pay attention to them. Behavioral tasks measure the capacity to accurately detect and perceive specific signals, including: (i) cardiac signals, for example *via* the heartbeat counting task (*Schandry, 1981*) or the heartbeat detection task (*Azevedo, Aglioti & Lenggenhager, 2016*; *Whitehead et al., 1977*); (ii) gastric signals, *via* the measurement of the perception of gastric contractions and feelings of stomach fullness (*Garfinkel et al., 2017*; *Herbert et al., 2012*; *Van Dyck et al., 2016*; *Whitehead & Drescher, 1980*); (iii) respiratory signals, *via* the pneumoception task (*Monti et al., 2020*) or more classical respiratory resistance tasks (*Garfinkel et al., 2016*; *Harver, Katkin & Bloch, 1993*; *Steptoe & Noll, 1997*); (iv) signals related to the movement and position of the body, *via* the threshold to detection of passive motion task (TTDPM), the joint position reproduction task (JPR), or the active movement extent discrimination apparatus (AMEDA), (*Lephart et al., 2002*; *Waddington & Adams, 1999*; *Weerakkody et al., 2008*); (v) visual signals, *via* the Body-Scaled Action Task (*Guardia et al., 2010*; *Valenzuela-Moguillansky, Reyes-Reyes & Gaete, 2017*); and (vi) tactile signals via the Somatic Signal Detection Task (*Durlik, Cardini & Tsakiris, 2014*).

In addition to behavioral tasks, self-report questionnaires have also been developed to assess different facets of conscious feelings of corporeal awareness, such as the tendency to notice or pay attention to body sensations and functions under normal and stressful conditions, and aspects related to the emotional and self-regulatory components of reading body signals. A limitation of existing self-report measures of corporeal awareness is that they either include items related exclusively to interoception, such as the Multidimensional Assessment of Interoceptive Awareness (MAIA-2) (*Mehling et al., 2018*) or bulk together across BSC related to different sensory domains, such as the Body Perception Questionnaire

(*Cabrera et al., 2018*; *Porges, 1993*), which measures interoception and proprioception. Therefore, to the best of our knowledge, there are no instruments available which evaluate only exteroceptive BSC. Indeed, there is evidence that although interoceptive and exteroceptive signals are highly interconnected (*Simmons et al., 2013*), they cannot be considered as the same construct and they are associated to at least partially different neural counterparts (*Farb, Segal & Anderson, 2013*; *Hurliman, Nagode & Pardo, 2005*). For instance, performance in the heartbeat counting task does not correlate with tasks measuring body awareness based on exteroceptive cues, such as tactile perception (*Crucianelli, Enmalm & Ehrsson, 2022*; *Durlik, Cardini & Tsakiris, 2014*; *Ferentzi et al., 2017*) or visual awareness of body size (*Vabba et al., 2022*; *Valenzuela-Moguillansky, Reyes-Reyes & Gaete, 2017*) and asking participants to focus on the body externally or internally has different effects on subsequent somatic perceptual decision-making, *i.e.*, exteroceptive body focus leads to more accurate detection of tactile stimulations in the SSDT, while a heartbeat perception task designed to increase interoceptive focus leads to a more liberal tactile decision criterion, with increased touch reports in the presence and absence of a tactile stimulation (*Mirams et al., 2012*). Given the importance of both interoceptive and exteroceptive signals in modulating body representation and higher-order cognitive and psychological processes, it is important to understand not only how these different types of signals are processed in the brain, but also how they are processed in subjective conscious experience. For example, self-reported exteroception may be interesting to study in itself as it has been linked to public self-consciousness (*Durlik, Cardini & Tsakiris, 2014*) which is the tendency to focus on aspects of the self that are visible to others (*Fenigstein, Scheier & Buss, 1975*). Furthermore, it could prove helpful in understanding and addressing dysfunctional body awareness and plasticity of the body schema in psychiatric conditions such as schizophrenia (*Graham et al., 2014*) and eating disorders (*Guardia et al., 2013*; *Provenzano et al., 2019*).

The current research aimed at developing and validating a new instrument, the Exteroceptive Body Awareness questionnaire (EBA-q), a self-report scale thought to isolate and measure aspects of BSC related to the processing of exteroceptive body-related signals (*e.g.*, vision, touch, smell). The authors made a list of items based on investigation of: (i) the literature concerning the theoretical constructs that they aimed to measure, and (ii) the already available instruments which could be related to exteroceptive BSC. Following approval of all items by the group and elimination of duplicate items, an initial pool of 33 items (10 reverse coded) was developed in Italian, which aimed at assessing visual, tactile, and olfactory awareness of the bodily self (see Supplementary Materials for a full list of the 33 original items). Visual items included questions related to the shape and size of the body, the coordination of the body in space, skin color, marks on the skin, and the ability to recognize oneself in mirrors, photos, and videos. The tactile items assessed awareness of tactile sensations on the body by various objects, such as the touch of other people, clothing and accessories, insects, food, and sweat. Finally, olfactory items assessed attention towards—and recognition of—the body based on odor (*e.g.*, bad smell). All the items were reviewed by the research team for clarity to avoid ambiguity, double negatives, and double barreling. Participants were asked to rate how much each expression—which

described specific sensations or behavior—was descriptive of them, on a 5-pont Likert scale ranging from 1 = "*not at all descriptive*" (in Italian "estremamente non caratteristica") to 5 = "*very descriptive*" (in Italian "estremamente caratteristica"). Thus, higher scores in the scale reflect higher awareness of bodily exteroceptive signals.

The purpose of the three validation studies was: (1) to select and retain items for the scale and to assess internal consistency and factor structure, (2) to confirm factor structure and consistency, and (3) to assess convergent and divergent validity based on comparison with other questionnaires measuring BSC, and behavioral measures of exteroception and interoception. As subjective measures of divergent validity, we included self-report questionnaires of interoceptive sensibility, specifically the Noticing and Body Listening subscales of the Multidimensional Assessment of Interoceptive Awareness (MAIA-2; *Mehling et al., 2018*), the Private Body Consciousness sub-scale (measuring the awareness of internal sensations) of the Body Consciousness Questionnaire (BCQ; *Miller, Murphy & Buss, 1981*), and the Private Self-Consciousness sub-scale (measuring aspects of the self that are not visible/observable) of the Self-Consciousness Scale (SCS; *Scheier & Carver, 1985*). All these measures reflect specific components of self-awareness that can be distinguished from the awareness of exteroceptive signals, and they reflect introspective modes of paying attention to the self, typically associated with interoceptive rather than exteroceptive BSC (*Ainley et al., 2013*; *Davies, 2005*; *Durlik, Cardini & Tsakiris, 2014*).

As subjective measures of convergent validity, we included subscales of general self-awareness questionnaires which focused on public body consciousness *i.e.,* the Public Body Consciousness subscale (measuring awareness of aspects of the body that could be observable) of the BCQ (*Miller, Murphy & Buss, 1981*) and the Public Self-Consciousness subscale (measuring aspects of the self that are overt) of the SCS (*Scheier & Carver, 1985*). Considering evidence for the link between exteroception and public self-consciousness (*Durlik, Cardini & Tsakiris, 2014*; *Fenigstein, Scheier & Buss, 1975*) we expected that public dimensions of self-awareness may be more linked to exteroceptive BSC and eventually to the EBA-q scores.

Furthermore, we investigated whether, similarly to interoception (*Garfinkel et al., 2015*), also exteroception is a multidimensional construct described by objective measures (the accuracy of being aware of one's own exteroceptive bodily signals, *i.e.,* exteroceptive accuracy), subjective beliefs about one's own body signals (the self-report awareness of one's own exteroceptive representation, *i.e.,* exteroceptive sensibility) and the correspondence between the two (exteroceptive meta-awareness). To do that we looked at the correlation between exteroceptive sensibility (the EBA-q total score), exteroceptive accuracy (performance in a behavioral task of visual body awareness *i.e.,* the Body-Scaled Action Task, BSAT, *Guardia et al., 2010*) and exteroceptive awareness (meta-awareness of performance in the BSAT). Finally, we also investigated whether participants would report similar abilities for each exteroceptive dimension with its interoceptive counterpart, namely interoceptive accuracy (performance in the Heartbeat Counting Task, HBC, *Schandry, 1981*), interoceptive sensibility (the Noticing and Body Listening sub-scales of the MAIA-2) and interoceptive awareness (meta-awareness of performance in the HCT).

## MATERIAL AND METHODS

### Participants

Participants were recruited *via* Prolific (http://www.prolific.co) or from a database of volunteers of the Social and Cognitive Neuroscience Laboratory (SCNLab). They were all fluent Italian speakers. The experimental procedures were approved by the Ethics Committee of the Department of Psychology (protocol n. 525/2018), Sapienza University of Rome and by the Ethics Committee of the IRCCS Fondazione Santa Lucia, Rome (protocols n. CE/PROG.659 and CE/PROG.865) and were in accordance with the 1964 Declaration of Helsinki. All participants read and signed the informed consent sheet before taking part in the study. All of them were naïve to the purpose of the research and received a monetary compensation for their time.

### Sample of Study 1

A sample of 371 volunteer participants (188 males; mean age = 30.73, standard deviation = 9.77) recruited via Prolific participated in Study 1. Participants completed online the original 33-item EBA-q using Survey Monkey (http://www.surveymonkey.com) alongside a battery of self-report questionnaires measuring different dimensions of BSC (see Measures for a detailed description). Sixty-nine participants were removed from the analysis, as they did not complete all the questionnaires, failed one or more attention checks, or completed the study in abnormally low time (less than half the average time of 15 min). The remaining sample consisted of 302 participants (167 males; mean age = 30.47, standard deviation = 9.62).

### Sample of Study 2

A sample of 184 participants (102 males, mean age = 25.8, standard deviation = 5.05) recruited through the SCNLab voluntary database for another experiment participated in Study 2 and completed the reduced 18-item EBA-q for confirmatory factor analysis. Responses to the questionnaire were collected online via Survey Monkey (http://www.surveymonkey.com).

### Sample of Study 3

A sample of 64 participants (26 males, mean age = 24.52, standard deviation = 4.98), recruited through the SCNLab voluntary database for another experiment, participated in Study 3, which aimed at characterizing the exteroceptive BSC construct. Participants completed the reduced 18-item EBA-q as well as other self-report questionnaires measuring different dimensions of BSC and underwent objective measures of exteroceptive and interoceptive BSC, namely the BSAT and HCT respectively (see Measures for detailed descriptions). Responses to the questionnaire were collected online via Survey Monkey (http://www.surveymonkey.com) whereas the behavioral tasks took place in the Department of Psychology at Sapienza University of Rome.

### Measures of convergent and divergent validity

The Public Body Consciousness ($\alpha = 0.65$) and Private Body Consciousness ($\alpha = 0.71$) sub-scales of the Body Consciousness Questionnaire (BCQ) (*Miller, Murphy & Buss, 1981*)
were used as measures of convergent and divergent validity respectively. The Public sub-scale includes seven items measuring attention to the social presentation of the body (*e.g.*, "When with others, I want my hands to be clean and look nice"). The Private sub-scale includes five items measuring attention towards internal body states (*e.g.*, "I am sensitive to internal bodily tensions"). Items are rated on a 5-point Likert scale ranging from $0 =$ "*extremely uncharacteristic*" to $4 =$ "*extremely characteristic*".

The Public Self-Consciousness ($\alpha = 0.77$) and Private Self-Consciousness ($\alpha = 0.73$) sub-scales of the Self-Consciousness Scale (SCS) (*Scheier & Carver, 1985*) were used as measures of convergent and divergent validity respectively. The public sub-scale includes seven items measuring attention to the self as visible to others (*e.g.*, "I care a lot about how I present myself to others"). The private sub-scale includes nine items measuring the tendency to be introspective and to attend to inner thoughts and feelings (*e.g.*, "I generally pay attention to my inner feelings"). Items are rated on a 4-point Likert scale ranging from $0 =$ "*not like me at all*" to $4 =$ "*a lot like me*".

The Noticing ($\alpha = 0.55$) and Body Listening ($\alpha = 0.69$) sub-scales of the Multidimensional Assessment of Interoceptive Awareness-2 questionnaire (MAIA-2) (*Mehling et al., 2012*) were used as measures of divergent validity. The sub-scales measure, respectively, awareness of uncomfortable, comfortable, and neutral body sensations (4 items, *e.g.*, "When I am tense, I notice where the tension is located in my body") and active listening to the body for insight (3 items, *e.g.*, "I listen for information from my body about my emotional state"). Items were rated on a 6-point Likert scale ranging from $0 =$ "*Never*" to $5 =$ "*Always*".

A modified version (as described by *Vabba et al., 2022*) of the Body-Scaled Action task (BSAT) (*Guardia et al., 2010*) was used as a behavioral measure of convergent validity. At the beginning of the task, the experimenter measured each participant's height and shoulder width. These parameters were entered in the *E-prime 2* (Psychology Software Tools, Pittsburgh, PA) script to create a series of personalized visual stimuli (*i.e.,* open doors) which were projected onto a white wall during the experiment. Participants stood at a 5-meter distance from the wall and had to judge whether their body could pass through the series of projected doors. The task was made up of two experimental blocks, Body Width and Body Height, each composed of four practice trials and 21 experimental trials. In the Body Width block participants observed a series of doors that varied in width based on the participant's shoulder width. Specifically, doors varied in steps of 1 cm up to 10 cm larger or thinner than participant's actual shoulder width. In this block, the door height was fixed at 20 cm taller than the participants' height. Participants observed each door with no time constraints and answered whether they could pass through the door without turning sideways, by selecting "Yes" of "No" from a keyboard in front of them. The Body Height block was similar to the Body Width block but in this case the doors varied in height based on participant's actual height, *i.e.*, they varied in steps of one cm up to 10 cm taller or shorter than the participant's actual height. In this block, the door width was fixed at 20 cm larger than the participant's actual shoulder width and they judged whether they could pass through the door without bending. The order of trials and blocks was randomized for all participants. To derive a measure of exteroceptive meta-awareness, at

the end of the task, participants were asked to judge their perceived accuracy in the task, using a visual-analogue scale (VAS) ranging from *0 = "not at all accurate"* to *"100 = completely accurate.*

The Heartbeat Counting Task (HCT) (*Schandry, 1981*) was used as a behavioral measure of divergent validity. In this task participants mentally counted their heartbeats in four trials of varying length (25, 35, 45, and 100 s) which were delimited by two acoustic tones delivered through headphones. Participants were asked not to take their pulse and not to give an estimation of the number of heartbeats, but report only heartbeats they truly perceived. Participants provided their response at the end of each trial using a keyboard. Their real heartbeats were recorded using a two-electrode portable custom-made ECG detector (MyHeart). E-Prime 2 software (Psychology Software Tools, Pittsburgh, PA) was used to handle instructions, trial order and response collection. To derive a measure of interoceptive awareness, at the end of the task, participants were asked to judge their perceived accuracy in the task, using a visual-analogue scale (VAS) ranging from *0 = "Not at all accurate"* to *"100 = Completely accurate".*

## Data analysis

All statistical analyses for the experiment were run *via* SPSS (IBM) and R (*R Core Development Team, 2013*, packages *psych, ltm, Hmisc.,* psycho, GPArotation, polycor, dplyr, EFA.dimensions, *lavaan, semPlots*). Bayesian correlations were calculated using the open-source software JASP Version 0.6.6.

## Exploratory factor analysis (Study 1)

Exploratory factor analysis was computed on the responses of participants from *Sample of Study 1* to the initial pool of 33 items. Initially, an r-matrix containing polychoric correlations between all pairs of items was performed to check the pattern of item associations and to identify items that were not meaningfully related to the others and that should be excluded. After checking that remaining items' internal reliability was acceptable (alpha values higher than 0.70 according to *Bland & Altman (1997)*) and assumptions were respected (KMO measure of sampling adequacy and Bartlett's test of sphericity), parallel analysis (*Horn, 1965*) was used to generate datasets based on permutations of the data and to suggest the number of factors for extraction. To select items for inclusion in the final scale, and to examine the psychometric properties of the questionnaire, exploratory factor analysis with principal axis factoring extraction and promax oblique rotation (as we expected items to share variance) was performed on the data, and factor loadings below 0.4 (*Brysbaert & Stevens, 2018*) were suppressed. The procedure was repeated until all items loaded significantly onto the factors.

## Confirmatory factor analysis (Study 2)

A confirmatory factor analysis on the reduced 18-item EBA-q was performed on *Sample of Study 2* to test whether, once accounting for the general latent variable (*i.e.,* exteroceptive body awareness), domain specific variables still accounted for additional observed variance which is external to the general latent variable. We compared fit statistics of a bi-factor model (with diagonally weighted least squares estimation for ordinal variables)

to alternative unidimensional, separate factors, and second-order models. The sample size was good for confirmatory factor analysis (*Klein, 2016*) and we required at least two of the following indices of fit to fall within the following standards (*Hu & Bentler, 1999*): *RMSEA* $\leq$ 0.06, *SRMR* $\leq$ 0.08, *CFI* $\geq$ 0.95, *TLI* $\geq$ 0.95, and the chi-square/df ratio less than 3 (*Klein, 2016*).

## Correlation analysis for testing convergent and divergent validity (Study 3)

### Correlations with subjective measures of body awareness

To examine convergent and divergent validity with other self-report measures, we combined responses from participants in *Sample of Study 1* and *Sample of Study 2*, calculated the total EBA-q scores as well as each of the sub-scales and performed Pearson's correlations between these measures and additional self-report measures of BSC, *i.e.,* sub-scales of the MAIA-2, BCQ, and SCS questionnaires. Where non-significant correlations were obtained, we also looked at the Bayes Factor to test the null hypotheses (using the open-source software JASP Version 0.14.1; 2020).

### Correlations with behavioral measures of body awareness

To further characterize the Exteroceptive Body Awareness construct, we performed Pearson's correlations between behavioral, self-reported, and meta-cognitive measures of exteroception and interoception in the responses of participants from *Sample of Study 3.* Specifically, we assessed the correlation between the total score of the 18-item EBA-q (exteroceptive sensibility) with performance in the modified version of the BSAT (exteroceptive accuracy) and meta-awareness of performance in the BSAT (exteroceptive awareness). Furthermore, we measured the relationship of exteroceptive measures with the behavioral, self-reported, and meta-cognitive measures of interoception *i.e.,* the Noticing and Body Listening sub-scales of the MAIA-2 questionnaire (interoceptive sensibility), performance in the HCT (interoceptive accuracy), and meta-awareness of performance in the HCT. Where non-significant correlations were obtained, we also looked at the Bayes Factor to test the null hypotheses (using the open-source software JASP Version 0.14.1; 2020).

Signal detection theory (*SDT*; *Stanislaw & Todorov, 1999*) was used on the scores in the BSAT. For each block, participant responses were categorized as *hits* (participants accurately judge that they can pass through the door), *misses* (participants judge that they cannot pass through the door when they can), *false alarms* (participants judge that they can pass through the door when they cannot), and *correct rejections* (participants accurately judge that they cannot pass through the door). *Hit rate* and *false alarm rate* were used to calculate, using the *psycho* the package on R-Studio software, the d' which was considered as the final measure of exteroceptive accuracy. Exteroceptive meta-awareness was calculated as the absolute difference between confidence judgement in overall performance (*i.e.,* scores in the final confidence VAS converted to decimal points) and the actual percentage of correct responses in the task, with the following formula, where scores closer to 1 indicated greater meta-awareness of the capacity to correctly detect whether the participant's body

fitted through the doors:

Exteroceptive awareness $= 1 - |$(final confidence VAS $-$ exteroceptive accuracy)$|$.

A Matlab (The MathWorks, Inc) custom script was used to identify and count the number of R-wave peaks on the ECG trace during the HCT, which was also visually inspected for artefacts. Interoceptive accuracy was calculated as the ratio of perceived to real heartbeats averaged across all trials, using the following formula:

$$\text{IAcc} = \frac{1}{4}\sum(1 - [|\text{recorded heartbeats} - \text{counted heartbeats}|]/\text{recorded heartbeats}).$$

Thus, scores closer to 1 indicated higher performance in the task. Interoceptive meta-awareness was calculated as the absolute difference between confidence judgement in overall performance (scores in the final VAS converted to decimal points) and the interoceptive accuracy score, using the following formula where scores closer to 1 indicated greater meta-awareness of the capacity to correctly detect heartbeats:

Interoceptive awareness $= 1 - |$(final confidence VAS $-$ interoceptive accuracy)$|$.

# RESULTS

## Exploratory factor analysis (Study 1)

To decide which items to eliminate we first inspected the r-matrix containing polychoric correlations between all 33 items presented to participants from *Sample of Study 1*, and eliminated seven items that had less than 2 correlations above the suggested cutoff of .3 (*Field, Miles & Field, 2012*). We next investigated factorability for the remaining 26 items: the KMO measure of sampling adequacy of 0.82 was meritorious (*Kaiser & Rice, 1974*) and Bartlett's test of sphericity was significant ($\chi = 2660.15$, $df = 325$, $p < .001$), indicating suitability of the data for exploratory factor analysis. We used parallel analysis (*Horn, 1965*) to decide the number of factors to extract, which suggested the extraction of four factors. Exploratory factor analysis with principal axis factoring extraction was performed on the dataset and promax oblique rotation was used as we assumed the factors to be correlated. The procedure was repeated until all items presented factor loadings $> .4$, leading to the elimination of eight extra items. Factorability analysis for the final 18 items indicated suitability of the data for exploratory factor analysis (determinant of coefficient $= 0.0031676 > 0.00001$, $KMO = 0.81$, *Bartlett's* $\chi = 1692.86$, $df = 153$, $p < .001$). Parallel analysis still suggested the extraction of four factors. All factor loadings were above the suggested 0.4 cut-off. Model fit was adequate ($\chi = 167.91$, $df = 87$, $p < .001$, $RMSA = 0.03$, $RMSEA = 0.055$, $TLI = 0.91$).

The final 18 items showed an internally consistent structure (mean $= 3.48$, standard deviation $= 0.48$, $\omega$ total $= 0.88$) and four factors cumulatively explained 44.26% of the variance. The first factor (mean $= 3.96$, standard deviation $= 0.55$, $\omega$ total $= 0.80$) accounted for 31.46% of the explained variance (13.92% of total variance), was composed of seven items and was called "Visuo-tactile body awareness" (*e.g.*, "I can immediately tell if a small insect sits on my skin".). The second factor (mean $= 4.03$, standard deviation

= 0.73, $\omega$ total = 0.84) accounted for 28.34% of the explained variance (12.54% of total variance), contained four items and was called "Spatial coordination" (*e.g.*, "When I walk, I often bump into people, because I don't realize how much space my body takes up"). The third factor (mean = 3.00, standard deviation = 0.82, $\omega$ total = 0.71) accounted for 21.48% of the explained variance (9.5% of total variance), contained four items and was called "Awareness of body changes" (*e.g.*, "I can immediately tell if my weight changes, even by a little"). The fourth factor (mean = 2.98, standard deviation = 0.83, $\omega$ total = 0.67) accounted for 18.71% of the explained variance (8.28% of total variance), contained three items and was called "Awareness of clothing fit" (*e.g.*, "I can immediately tell if an item of clothing will fit me, even before trying it on"). Factor loadings for all items constituting the 18-item questionnaire are listed in Table 1. Correlations between the four factors are presented in Table 2.

## Confirmatory factor analysis (Study 2)

We fit a bifactor model to the data, to assess if and how much domain specific factors accounted for additional variance which was not explained by the general factor of exteroceptive body awareness. Following the standard procedure, we compared the fit of this model to that of a second-order model (in which the lower-order factors are substantially correlated with one another and there is a single higher-order factor that accounts for these correlations), a unidimensional model and a separate factors model. Fit statistics for the bifactor model (with diagonally weighted least squares estimation for ordinal variables) as well as alternative unidimensional, separate factors, and second-order models are presented in Table 3.

The bifactor model was the only model to present adequate fit ($\chi$ = 292.98, $df$ = 117, $p < .001$, $SRMR$ = 0.08, $RMSEA$ = 0.08, $CFI$ = 0.95, $TLI$ = 0.93). All variances had a positive sign, and R-squares were under 1. Model comparison showed that the bifactor model presented better fit compared to the separate factors model ($\chi$ difference = 57.29, $df$ = 12, $p < .001$), compared to the second order model ($\chi$ difference = 153.98, $df$ = 14, $p < .001$), and compared to the unidimensional model ($\chi$ difference = 535.75, $df$ = 18, $p < .001$).

Once accounting for variance explained by the general factor of exteroceptive body awareness (mean = 3.43, standard deviation = 0.52, $\omega$ total = 0.87), additional variance was explained by the factors: Spatial coordination (mean = 3.90, standard deviation = 0.74, $\omega$ total = 0.79), Awareness of body changes (mean = 3.02 standard deviation = 0.90, $\omega$ total = 1.07), and Awareness of clothing fit (mean = 2.84, standard deviation = 0.90, $\omega$ total = 0.91), suggesting part of the variance in these three factors may be explained by other factors external to exteroceptive body awareness. However, the factor Visuo-tactile body awareness (mean = 3.94, standard deviation = 0.57, $\omega$ total = 0.68), did not clearly explain additional variance, as evidenced by partially non-significant estimate values for the items related to these factors (see Fig. 1 for item loadings). This suggests that the variance explained by Visuo-tactile body awareness is mostly caused by the same factors influencing the general latent variable of Exteroceptive Body Awareness.

**Table 1** Factor loading, mean, and standard deviation for the 18 items composing the Exteroceptive Body Awareness questionnaire. All relevant items were reverse coded before exploratory factor analysis.

| Item | | F1 | F2 | F3 | F4 | M (SD) |
|---|---|---|---|---|---|---|
| 1 | I can immediately tell if I will be able to reach an object on a high shelf, without using a support. | 0.55 | | | | 3.92 (0.95) |
| 2 | I understand immediately if I will be able to pass under a low ceiling, without bending over. | 0.46 | | | | 3.75 (0.89) |
| 3 | I can immediately tell if and how much I need to rotate my body to pass through a passage. | 0.57 | | | | 4.00 (0.90) |
| 4 | When someone touches me, I can precisely tell in which spot I was touched. | 0.74 | | | | 4.18 (0.85) |
| 5 | I can immediately tell if a small insect sits on my skin. | 0.64 | | | | 3.62 (1.03) |
| 6 | I can immediately tell when I start sweating, even if just a little bit. | 0.43 | | | | 3.95 (0.93) |
| 7 | When I end up in a puddle, I immediately realize if my foot gets wet. | 0.60 | | | | 4.29 (0.74) |
| 8 | I often hit my head because I underestimate my height (RC). | | 0.52 | | | 4.20 (0.95) |
| 9 | I believe I have good awareness of my body in space. | | 0.47 | | | 3.85 (0.83) |
| 10 | When I walk, I often bump into people, because I don't realize how much space my body takes up (RC). | | 0.86 | | | 4.22 (0.96) |
| 11 | I often bump into furniture, objects, and doors, even in familiar environments, because I don't realize how much space my body takes up (RC). | | 0.96 | | | 3.83 (1.14) |
| 12 | I can immediately tell if my weight changes, even by a little. | | | 0.66 | | 3.04 (1.17) |
| 13 | When I spend time in the sun, I am very aware of the change in my skin tone, even if it is small. | | | 0.52 | | 3.41 (1.19) |
| 14 | I can immediately tell if marks appear on my skin. | | | 0.54 | | 3.07 (1.13) |
| 15 | I can immediately tell if my moles change size, even if only by a little. | | | 0.67 | | 2. 48 (1.10) |
| 16 | I can immediately tell if an item of clothing will fit me, even before trying it on. | | | | 0.64 | 3.13 (1.09) |
| 17 | I can immediately tell if a belt will fit me, or if I need to add holes to it because it is too small or too big. | | | | 0.53 | 2.93 (1.06) |
| 18 | I can immediately tell if a pair of shoes will fit me, even before trying it on. | | | | 0.78 | 2.77 (1.07) |

**Notes.**
M, mean; SD, standard deviation; F1, F2, F3, F4, Factor 1,2,3,4.

**Table 2** Promax rotation factor correlations.

| Factor | F1 | F2 | F3 | F4 |
|---|---|---|---|---|
| Visuo-tactile awareness (F1) | 1.00 | 0.41 | 0.35 | 0.55 |
| Spatial coordination (F2) | 0.51 | 1.00 | −0.05 | 0.25 |
| Awareness of body changes (F3) | 0.35 | −0.02 | 1.00 | 0.38 |
| Awareness of clothing fit (F4) | 0.55 | 0.28 | −0.38 | 1.00 |

**Notes.**
M, mean; SD, standard deviation; F1, F2, F3, F4, Factor 1,2,3,4.

**Table 3 Goodness of fit indices for the four models tested for confirmatory factor analysis.**

| Model | Chi-square | SRMR | RMSEA | CFI | TLI |
|---|---|---|---|---|---|
| Unidimensional | $\chi = 906.14, df = 135, p < .001$ | 0.13 | 0.15 | 0.77 | 0.74 |
| Separate factors | $\chi = 375.09, df = 129, p < .001$ | 0.09 | 0.09 | 0.93 | 0.91 |
| Second-order | $\chi = 483.78, df = 131, p < .001$ | 0.10 | 0.11 | 0.89 | 0.87 |
| Bifactor | $\chi = 292.98, df = 117, p < .001$ | 0.08 | 0.08 | 0.95 | 0.93 |

**Notes.**
$N = 248$.
SRMR, standardized root mean square residual; RMSEA, root mean square error of approximation; CFI, Comparative Fit Index; TLI, Tucker Lewis Index.

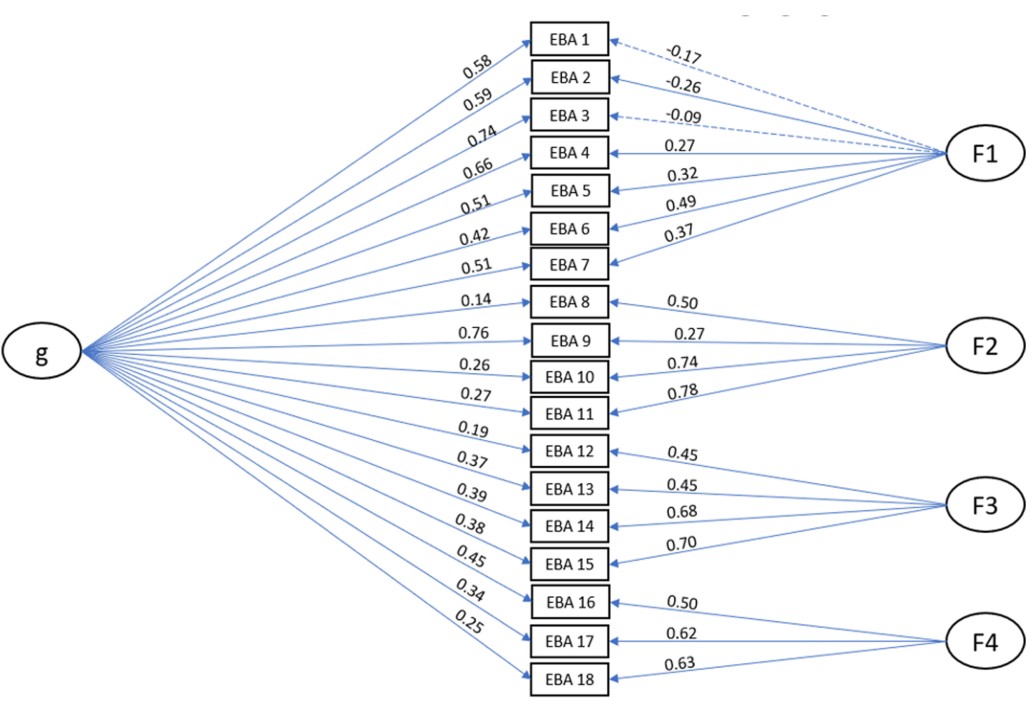

**Figure 1 Bi-factor model resulting from confirmatory factor analysis for the structure of the Extero-ceptive Body Awareness (EBA) questionnaire.** Measured variables (questionnaire items) are represented by rectangles. Latent variables are represented by ellipses. g, the general construct of exteroceptive body awareness; F1, Visuo-tactile body awareness; F2, Body coordination; F3, Awareness of body changes; F4, Awareness of clothing fit. Values on the lines indicate the b estimates. Dotted lines indicate estimates did not reach significance at the $p < .05$ level.

## Correlation analysis for testing convergent and divergent validity (Study 3)
### Correlations with subjective measures of body awareness

To assess convergent and divergent validity with other self-report measures assessing similar or different aspects of body awareness, we examined the correlation between the EBA-q total scores and sub-scale scores, with self-reported measures of private and public BSC and with each separate dimension of interoceptive sensibility. To do that and rely on a larger sample size, we combined responses from participants in *Sample of Study 1* and *Sample of Study*

**Table 4 Means, standard deviations, and correlations between scales/behavioural measures for convergent and divergent validity with the total of the EBA questionnaire and each sub-scale.**

| Test | M (SD) | R | | | | |
|---|---|---|---|---|---|---|
| | | EBA total | Visuo-tactile body awareness | Body coordination | Awareness of body changes | Awareness of clothing fit |
| Noticing (MAIA-2) | 4.22 (0.81) | 0.320[*] | 0.269[*] | 0.052 | 0.274[*] | 0.245[*] |
| Body Listening (MAIA-2) | 3.89 (1.00) | 0.286[*] | 0.136[*] | 0.023 | 0.301[*] | 0.253[*] |
| Private Body Consciousness (BCQ) | 13.52 (4.55) | 0.269[*] | 0.224[*] | 0.032 | 0.296[*] | 0.154[*] |
| Public Body Consciousness (BCQ) | 21.05 (3.81) | 0.375[*] | 0.288[*] | 0.066 | 0.417[*] | 0.207[*] |
| Private Self-Consciousness (SCS) | 18.37 (4.26) | 0.208[*] | 0.268[*] | −0.068 | 0.261[*] | 0.111[*] |
| Public Self-Consciousness (SCS) | 12.88 (4.12) | 0.150[*] | 0.147[*] | −0.087 | 0.247[*] | 0.084 |

Notes.

M, mean; SD, standard deviation; R, Pearson's correlations; N, number of subjects = 364.

*indicates significance below the Bonferroni cutoff for multiple comparisons ($p = .003$).

*2.* The obtained Pearson's correlations are reported in Table 4, as well as means, standard deviations, and McDonalds $\omega$ total values for each measured sub-scale. The Bonferroni cutoff for significance for multiple comparisons was set as $p = .003$. As predicted, and supporting convergent validity, EBA-q total scores correlated significantly with Public Body Consciousness (BCQ) ($r = .375, p < .001$) and Public Self-Consciousness (SCS) ($r = .150, p = .004$). However, contrarily to our prediction, the EBA-q total scores also correlated significantly with measures of Private Body Consciousness ($r = .269, p < .001$) and Private self-consciousness ($r = .208, p < .001$) as well as with specific dimensions of interoception from the MAIA-2 questionnaire, specifically Noticing ($r = .320, p < .001$) and Body Listening ($r = .286, p = .001$).

### Correlations with behavioral measures of body awareness

To investigate if, similarly to the tridimensional construct of interoception proposed by *Garfinkel et al. (2015)*, also exteroception would present separate dimensions of accuracy, sensibility, and awareness, in Study 3, we correlated the EBA-q with performance on the Body Scale Action Task (BSAT), as well as an index of exteroceptive meta-awareness.

Specifically, the EBA-q total score was considered a measure of exteroceptive sensibility (mean = 3.47, standard deviation = 0.47), the individual $d'$ for performance in the BSAT was considered a measure of exteroceptive accuracy (mean = 1.76, standard deviation = 0.58), and the absolute ratio between confidence in performance reported in a 0–100 VAS at the end of the BSAT and actual percentage of correct responses was considered a meta-awareness measure of exteroceptive awareness (mean = 0.80, standard deviation = 0.15). As for interoception, exteroceptive sensibility did not correlate with exteroceptive accuracy ($r = 0.09, p = 0.48, BF = 0.198$) or with exteroceptive meta-awareness ($r = -0.03; p = .81, BF = 0.160$), and exteroceptive accuracy and awareness also did not correlate ($r = -0.02; p = .897, BF = 0.157$). This result suggests that, like interoception, exteroceptive bodily awareness is a multidimensional construct.

We then calculated the three dimensions of interoception proposed by *Garfinkel et al. (2015)* and explored their associations in our sample. Specifically, the Noticing sub-scale of the MAIA-2 questionnaire was used as a measure of interoceptive sensibility (mean =

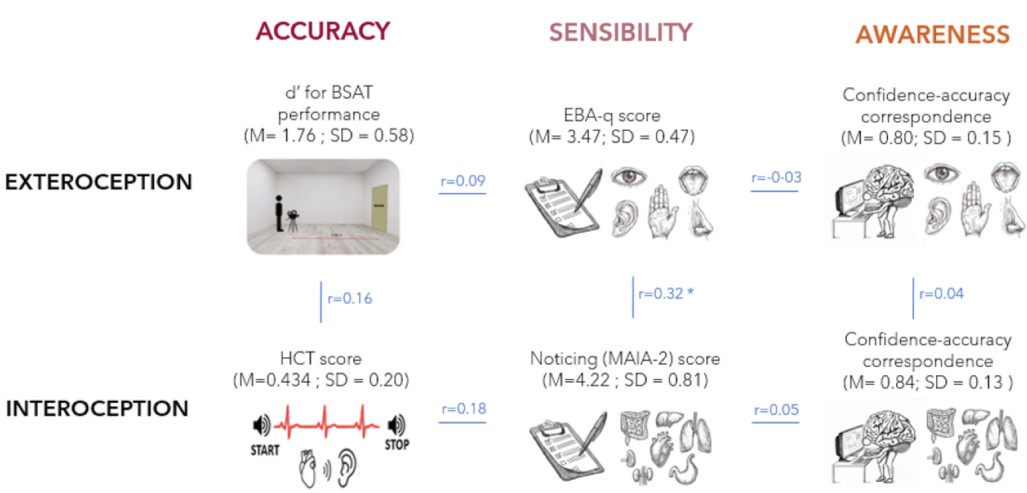

**Figure 2  Diagram representing all measured variables of exteroception and interoception (accuracy, sensibility, and awareness) and their relationship.** BSAT, Body-Scaled Action Task; HCT, Heartbeat Counting Task. * indicates significance at $p < .001$.

4.22, standard deviation = 0.81), the score in the heartbeat counting task (HCT) was used as a measure of interoceptive accuracy (mean = 0.434, standard deviation = 0.20), and the absolute ratio between confidence in performance reported in a 0–100 VAS at end of the HCT and interoceptive accuracy was used as a measure of interoceptive awareness (mean = 0.84, standard deviation = 0.13). Interoceptive accuracy did not correlate with interoceptive sensibility ($r = 0.182$; $p = .156$, $BF = 0.423$) or with interoceptive awareness ($r = -0.166$; $p = 0192$, $BF = 0.361$) and interoceptive sensibility and meta-awareness also did not correlate ($r = 0.047$; $p = .718$, $BF = 0.170$).

We finally tested for associations across interoceptive and exteroceptive domains. We found that interoceptive accuracy did not correlate with exteroceptive accuracy ($r = 0.16$, $p = 0.22$, $BF = 0.327$) nor did interoceptive awareness correlate with exteroceptive awareness ($r = 0.04$, $p = .776$, BF = 0.164). However, there was a significant correlation between interoceptive and exteroceptive sensibility ($r = 0.320$, $p < .001$). These results suggest that while interoceptive and exteroceptive accuracy are distinct abilities relying on distinct processes, at the subjective level there is an association between how aware we are about our interoceptive and exteroceptive body representation.

A graphical representation of the different dimensions of interoception and exteroception can be found in Fig. 2.

## DISCUSSION

Pre-existing self-report measures of body awareness mainly focused on signals coming from inside the body, namely from the viscera (interoceptive) and muscles (proprioceptive). However, previous research examining the role of bodily self-consciousness on cognition and behavior suggests that the processing of both interoceptive and exteroceptive signals crucially shapes Bodily Self Consciousness (BSC), and relies on highly interconnected

but distinct processes, with dissociable neural counterparts (*Farb, Segal & Anderson, 2013*; *Hurliman, Nagode & Pardo, 2005*; *Simmons et al., 2013*; *Valenzuela-Moguillansky, Reyes-Reyes & Gaete, 2017*). Based on this theoretical and evidence-based distinction and on the fact that self-reported measures of exteroceptive bodily awareness are missing, we developed a questionnaire aimed at isolating features of bodily self-consciousness specifically related to the awareness of exteroceptive body signals. Our results provide preliminary evidence for the reliability and validity of the Exteroceptive Body Awareness questionnaire (EBA-q) as an instrument for measuring participants' self-reported capacity to correctly make appraisals about the state of (or changes in) the body, which are based on the processing of exteroceptive signals.

## Exploratory factor analysis

Exploratory factor analysis revealed a four-factor structure. The first factor was called *Visuo-tactile body awareness* and included seven items measuring participants' self-reported visual awareness of the body dimensions in space (*e.g.*, assessing whether the body can pass through tight passages or under low ceilings) and precision in detecting tactile stimuli on the body (*e.g.*, other people's touch, insects, sweat, and puddles). The second factor was called *Body coordination* and included four items measuring participants' level of clumsiness and spatial awareness when interacting with objects in space (*e.g.*, bumping into people and furniture). The third factor was called *Awareness of body changes* and included four items measuring awareness of changes in the body (*e.g.*, weight changes, appearance of marks and moles, tanning). The fourth factor was called *Awareness of clothing fit* and contained three items measuring awareness of body size in relation to clothing (*e.g.*, clothing items, belts, shoes). The total cumulative variance explained by the factor structure was 44.26% indicating that the remaining variance in the questionnaire items was not captured by the described factors.

We suggest that unexplained variance in our general construct of exteroceptive bodily awareness may be caused by external factors contributing to bodily self-consciousness not directly measured by the EBA-q questionnaire. In fact, the main aim of the EBA-q was to measure our capacity to be aware of exteroceptive bodily cues. In line with this idea, our experience of the body is not straightforward—it is built upon the integration of moment by moment perceptual cues coming from inside and outside the body, as well as cognitive and emotional factors and our existing implicit and explicit models of body representation, created thanks to prior experience, which help us to make sense of sensory input (*Blanke, 2012*).

In fact, bodily self-consciousness, has been described by researchers as constituted by different components, namely the *body image*, conceived as a conscious long-term perception and experience of the physical self in terms of the size, shape, and physical representation of our body, the *body schema*, conceived as a dynamic model of the body posture underlying skilled actions, and the *superficial schema*, mediating localization of stimuli on the body surface (*Longo, 2015*). Some of these domains are likely to be influenced by cognitive and affective factors besides the perceptual ones. For example, the cognitive-behavioral model proposed by Cash (see *Thompson & Schaefer, 2019*)

suggests that body image is a result of historical events (*i.e.,* past events that predispose thoughts and feelings about body image, including cultural values and standards of physical appearance, interactions with other, physical characteristics and personality) and proximal circumstances (*i.e.,* perceptual, and emotional inferences about the current state of the body). Thus, it is possible that items, at this time missing, measuring those higher-level factors would have explained the remaining variance.

## Confirmatory factor analysis

Considering the low correlations between factors, the low cumulative variance explained by the factors resulting from the exploratory factor analysis, and the high correlation between each factor and the scale total, a bifactor model was deemed the most appropriate fit for the data. In a bifactor model, a general factor accounts for the commonality of all items, and domain specific factors account for additional and unique influence of the specific domain which is not described by the general factor (*Chen et al., 2020*). We compared this model *via* goodness-of-fit measures to a second-order model (in which the lower-order factors are substantially correlated with one another and there is a single higher-order factor that accounts for these correlations), a unidimensional model and a separate factors model. Our findings confirmed a bifactor model to be the best—as well as the only adequate—model fit for the data.

Once accounting for variance explained by the general factor, additional variance was explained by *Spatial coordination*, *Awareness of body changes*, and *Awareness of clothing fit*. These findings suggest that the items included in these subscales may describe other domains of exteroceptive bodily awareness. Specifically, the Spatial coordination subscale may describe exteroceptive body awareness related to motor control and spatial awareness. This is in line with the idea that Exteroceptive body awareness refers to the knowledge of having a body in relation to space and movement since this body representation comes from the integration of multimodal exteroceptive signals (*e.g.*, vision, sound, touch), vestibular and proprioceptive systems, and voluntary motor systems (*Tsakiris, 2017*). Awareness of clothing fit and awareness of body changes may describe aspects of Exteroceptive Body Awareness related to the awareness of this plastic representation. Indeed, body image concerns can lead to malleable and distorted body representations (*Beckmann et al., 2021*; *Guardia et al., 2010*). These originate in the brain from a complex time to time integration of bottom-up sensory signals that vary across time (*e.g.*, changes in weight or skin tone, *etc.*) and that are integrated with top-down cognitive processes in order to build an unitary concept of the sense of self (*Villani, Tsakiris & Azevedo, 2019*).

Finally, the factor *visuo-tactile awareness* did not clearly explain additional variance, as evidenced by partially non-significant estimate values for the items related to the factor, suggesting that the majority of variance in these items was explained by the general factor of exteroceptive body awareness.

Given these results, we suggest that the questionnaire may be a valuable instrument for researchers interested in both –the general construct of exteroceptive body awareness and the domain specific factors.

## Correlation analysis for testing convergent and divergent validity

Confirming convergent validity, we found that the 18-item EBA-q correlated significantly with public measures of body and self-consciousness, *i.e.,* Public Body Consciousness (BCQ) and Public Self-Consciousness (SCS). However, the EBA-q did not significantly correlate with the behavioral measure of visual body awareness (the Body-Scaled Action Task, BSAT) in which participants had to guess whether they could fit through a series of projected doors varying in height and width) nor with the meta-awareness score related to performance in the task.

Contrarily to our hypothesis, for divergent validity the 18-item EBA-q showed positive although moderate (*Dancey & Reidy, 2011*) correlations with self-report measures of interoceptive sensibility (Noticing and Listening sub-scales of the MAIA-2) and to private measures of body and self-consciousness *i.e.,* Private Body Consciousness (BCQ) and Private Self-Consciousness (SCS). With respect to the behavioral measure of interoceptive accuracy (the heartbeat counting task, HCT) we did not find a significant correlation although the Bayes Factor for this correlation suggested only anecdotal evidence. Finally, in line with previous research (*Durlik, Cardini & Tsakiris, 2014*; *Valenzuela-Moguillansky, Reyes-Reyes & Gaete, 2017*), objective measures of exteroceptive and interoceptive accuracy did not correlate.

## Characterizing the exteroceptive body awareness construct

We interpreted these findings as providing preliminary evidence that, exteroception is a construct composed by multiple dimensions, which vary from self-report to objective assessment. This mirrors the distinction highlighted by *Garfinkel et al. (2015)* in the construct of interoception, which the authors described as composed of at least three components of interoceptive awareness namely: interoceptive accuracy, *i.e.,* the capacity to accurately detect internal body signals in behavioral tasks (*e.g.,* the HCT); interoceptive sensibility, *i.e.,* the general tendency to focus on or pay attention to internal signals, measured through self-report questionnaires, and interoceptive awareness, *i.e.,* the meta-awareness of the capacity to accurately detect interoceptive signals, measured through accuracy-awareness correspondence (confidence) ratings following behavioral tasks. Typically, these dimensions of interoception do not correlate. The results of our study suggest that a similar distinction between accuracy (performance in the BSAT), sensibility (the EBA-q), and awareness (confidence-accuracy correspondence for the BSAT) extends also to exteroceptive body awareness.

It is worth noticing that contrarily to our prediction, there was a significant, although moderate (*Dancey & Reidy, 2011*) positive correlation between the EBA-q and measures of interoceptive sensibility and private self-consciousness (*i.e.,* the Noticing and Body Listening sub-scales of the MAIA-2 and measures of Private Body Consciousness and Private Self-Consciousness). This finding suggests that, while behavioral measures of interoceptive and exteroceptive accuracy do not correlate, the self-reported tendency to pay attention to the body and to signals arising from different sensory modalities (interoceptive and exteroceptive sensitivity) may share similar processes. Indeed it is possible that while the ability to map the status of different sensory modalities (*e.g.,* visual

exteroception and cardiac interoception) relies on dedicated detection processes that may differentially influence cognitive functions and behaviors (*Crucianelli, Enmalm & Ehrsson, 2022*; *Durlik, Cardini & Tsakiris, 2014*; *Ferentzi et al., 2017*; *Mirams et al., 2012*; *Valenzuela-Moguillansky, Reyes-Reyes & Gaete, 2017*), our overall tendency to pay attention to different bodily signals relies on a more general ability. Thus, to have a complete picture of the bodily self, it is fundamental to explore interoception, proprioception, and exteroception, both at the behavioral and subjective level and how they are integrated. Indeed, there are circumstances in which the same bodily signals overlap and compose different patterns that we interpret differently. For example, visual and proprioceptive signals are both fundamental for perceiving movement and body position in space, and the sensation of chest wall muscles is relevant to breathing, or to detect cardiac pain (see *Critchley & Garfinkel, 2018*).

## LIMITATIONS

Although this study offers preliminary evidence of the EBA-q's internal reliability and convergent validity with other measures of body awareness, it presents some limitations. All questions in the survey reflect participants' belief in their capacity to make appraisals about the body based on exteroceptive cues. There may be additional aspects of exteroceptive awareness that are not captured by the questionnaire, such as the tendency to regulate attention towards the exteroceptive body and to worry about it (similarly to the Attention Regulation and Not-Worrying sub-scales of the MAIA-2). Furthermore, items related to body odor were eliminated following exploratory factor analysis, possibly due to the small and redundant number of related items included in the scale. Due to the power that olfactory signals might have in shaping one's own representation (*Lundström & Olsson, 2010*) future attempts to add items related to olfactory awareness of the body may provide additional useful information regarding exteroceptive body awareness. Furthermore, while evidence was provided for convergent validity with other questionnaires measuring aspects of body awareness, we did not examine test-retest validity and the questionnaire did not present adequate divergent validity. Future research could test the conceptual relation between the EBA-q and social desirability or body image, for example by investigating body image concerns in questionnaires such as the as the Body Shape Questionnaire (*BSQ*) (*Cooper et al., 1987*) and the Body Uneasiness Test (*Cuzzolaro et al., 2006*) both measuring body image corncerns.

## CONCLUSION

Overall, we think that the EBA-q offers a fast and reliable measurement of participant's subjective awareness of the body based on the processing of exteroceptive signals. The entire 18-item scale—as well as each of the sub-scales—can be used in studies examining the role of bodily self-consciousness in cognitive functions and behaviors, and by researchers who aim to integrate measures of interoceptive and exteroceptive signal processing. Moreover, it may be also useful in clinical research in combination with other instruments to assess dysfunctional body awareness, for example in anorexic persons.

### Funding
The study was supported by European Research Council (ERC) Advanced Grant (eHONESTY, Prot. 789058). The funders had no role in study design, data collection and analysis, decision to publish, or preparation of the manuscript.

### Grant Disclosures
The following grant information was disclosed by the authors:
European Research Council (ERC) Advanced Grant: eHONESTY, Prot. 789058.

### Competing Interests
The authors declare there are no competing interests.

### Author Contributions
- Alisha Vabba conceived and designed the experiments, performed the experiments, analyzed the data, prepared figures and/or tables, authored or reviewed drafts of the article, and approved the final draft.
- Giuseppina Porciello conceived and designed the experiments, analyzed the data, authored or reviewed drafts of the article, and approved the final draft.
- Maria Serena Panasiti conceived and designed the experiments, analyzed the data, authored or reviewed drafts of the article, and approved the final draft.
- Salvatore Maria Aglioti conceived and designed the experiments, analyzed the data, authored or reviewed drafts of the article, and approved the final draft.

### Human Ethics
The following information was supplied relating to ethical approvals (*i.e.,* approving body and any reference numbers):

Approval to carry out the studies was granted by the Ethics Committee of the IRCCS Fondazione Santa Lucia (Rome) (protocol CE/PROG.659 and prot. CE/PROG.865) and the Ethics Committee of the Department of Psychology, Sapienza University of Rome (prot. n. 525/2018).

### Data Availability
The anonymized datasets analysed in the current study are available at OSF: Vabba, Alisha, Maria S Panasiti, Giuseppina Porciello, and Salvatore M Aglioti. 2023. "eH_WP3_L1." OSF. May 10.osf.io/mdarz.

### Supplemental Information
Supplemental information for this article can be found online at http://dx.doi.org/10.7717/peerj.15382#supplemental-information.

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
