# Peer review of "Development and validation of the Exteroceptive Body Awareness (EBA-q) questionnaire"

_PeerJ, doi:10.7717/peerj.15382_

## Round 0.1 · original submission · Minor Revisions

The two expert reviews offer detailed and insightful feedback regarding your manuscript, for which I am incredibly grateful. As you can see from their feedback, both reviewers have offered a generally positive assessment of your manuscript and felt that it is well-positioned to make a substantial and valuable contribution to the literature. I agree with this positive evaluation. Both expert reviews are precise and clear and have offered several concrete suggestions for improving your manuscript. I will, therefore, not reiterate all their points here. But I ask you to consider each of them when revising the manuscript to make it eligible for publication in PeerJ.

Reviewer 1 ·

Basic reporting

Basic Reporting was satisfactory.
Since this is a tool that hopefully will be useful to several researchers than it should be important that the data are shared, so I would encourage to make them already available in a repository rather than leaving them on request.

Experimental design

In this paper, the Authors introduce and validate a novel self-report questionnaire to measure the construct of exteroceptive body awareness. They highlight that a questionnaire capturing such measure alone is currently lacking and they aim to fill such gap.
Overall, I think this is an interesting and potentially useful tool, although several aspects coming from the results of this paper suggest that exteroceptive body awareness is rather not independent/nor separate by interoceptive body awareness (i.e. the correlational analyses of the EBA questionnaire with other interoception-related questionnaires) and that perhaps scores resulting from this questionnaires will not easily be able to correlate with behavioural accuracy measures.

I have below few suggestions that may improve the clarity of the results presented in the paper and the relevance of the research question raised up here:

Introduction
I thought that some sessions could be expanded:

-Page 4 – when you mention the Body Perception Questionnaire you say that measures “interoceptive and proprioceptive awareness”. Here, you seem to interchangeably consider the term proprioceptive with the term “exteroceptive”. I think is important in the intro to highlight in one paragraph what are the similarities and differences of exteroception with body proprioception – Is proprioception more “sensory” and exteroception more related to awareness? Does exteroception include also proprioception but not the opposite? It would be nice to understand this better.
Generally, here every comparison of such scale is to interoception but there is very little in the paper (not even a paragraph) talking how this measure might relate to sensori-motor, proprioceptive processes. I suppose there are plenty of other scales in that field, and such questionnaire might be useful also for research in those domains in principle.

After in the same page you say – “interoceptive and exteroceptive attention have opposite effects on subsequent somatic perceptual decision making” - > in which way? Would be nice to provide some more explanations to readers that not necessarily know all the interoception literature. Also opposite effects on somatic perceptual decision making may suggest that rather than being measured separately interoceptive and exteroceptive awareness should sistematically be measured together.

The name EBA is nice and provocative and of course relates to fMRI regions related to body perception. My personal opinion is that such name, commonly associated to a brain region related to body perception, may not help your questionnaire to stand-out and be well noticed. This is a personal preference of the Authors but perhaps I would rather change the acronym – such as: EBAq maybe? Just a suggestion.

Page 5: Could you please explain, maybe in the methods, or in the discussions why you do use an initial pool of 33 items? I think this should be justified because this number seem rather little?

The labels “Very Characteristic” “very uncharacteristic” – Could you please explain how you came up with such labels? They sound a bit strange in English, although maybe is not the case in Italian. (perhaps the translation of these in Italian should be visible in the paper).

Results

I think it would be VERY beneficial if you could integrate the analysis pipeline, while illustrating the results - making the result section a bit more discoursive and easier to follow (for some reason the submitted word file had the methods at the end – and perhaps this did not help because I went all the time back and forth to understand the results!!). Either way, especially because this paper is methodological and meant for people to make use of this scale, I suggest you to make them more readable, explaining every step while presenting its related statitics.

- 44% of explained variance seem rather little to me – how did you end up choosing 4 factors? Would it be possible to see the scree plot? In the limitations later in the paper you should justify or discuss why it may reached only 44% of the total variance, and what it is maybe missing?

- Maybe is uncommon, but I think it would be clearer if you specified the variance explained by each of the factors not relative to 100% but rather relative to the 44% that these factors manage to explain.

-The first dimension (which seem to explain most variance) seem to capture visuo- tactile body awareness. However, I found quite funny (at least in the english translation) how much was related to confidence (all of the questions include a “I can immediately/precisely tell”) much more than in other dimensions (except of the fourth clothes-related one). Did you think that this could be a factor? This interpretation may fit also later analysis in which this dimension alone does not help much explaining variance (additional to the EBA).

Results exp 2

Again I would do the same, making the results more discoursive integrating it with the analysis pipeline.

2.3.1
I think is not really useful to have a result section that simply points to a table – is not very informative as it is. Also it is quite surprising to see that interoception is positively associated with exteroception – This needs more motivation. It would really help also to have the methods related to those questionnaires already briefly explained. As it is is uninformative this section.

Validity of the findings

Discussions

3.1 – Here rather than discussing you only repeat the results, maybe is a good place to explain why you chose 4 factors even if they explain less than half variance?

3.2 Page 14 – I found the explanations of the bifactor models very helpful and quite clear – I would move a summary of this in the results section.

Would be nice if here you discuss possible alternative explanations of the factor no. 1 (see confidence, or any other possible interpretation you have) since this factor alone did not help much explaining additional variance.

One aspect is that interoceptive accuracy dissociates from interoceptive awareness, (e.g. negative correlation) – while less interesting is if the two measures do not correlate. Same thing for the exteroceptive accuracy. To my knowledge these questionnaires are usually helpful to relate them to actual behavioural performance. Don’t you think that this complete lack of relationship is possibly a reason to discourage the use of this questionnaires? Perhaps it would be informative to know (if you have them) how other interoceptive awareness scales correlate with these accuracy measures?

Page 16 just before the limitations paragraph: the sentence is not clear – not sure if you mean that interoception and exteroceptions both share commonalities related to body awareness (1 dimension) or they are indeed distinct dimensions of body awareness. I think based on your results that the former interpretation is more likely. This seem a key point/conclusion to the paper, would be good if stated more clearly!

Conclusion
“for researchers who aim to distinguish between interoceptive and exteroceptive signal processing” – Not sure that the questionnaire would necessarily allow that to happen given the high correlations, perhaps you could soften the claim and say that is for researchers who aim to “integrate” interoceptive measures with those exteroceptive.

Annotated reviews are not available for download in order to protect the identity of reviewers who chose to remain anonymous.

Reviewer 2 ·

Basic reporting

no comment

Experimental design

no comment

Validity of the findings

no comment

Additional comments

In this paper, Vabba and colleagues presented three studies they performed to develop and validate a questionnaire (the Exteroceptive Body Awareness EBA questionnaire). The questionnaire aims at assessing bodily self-consciousness from exteroceptive bodily information. The paper is well-written and the topic is relevant. The questionnaire can actually be a method for filling some gaps in the investigation of body representation. I have no major comments, but I would suggest clarifying the following points/ addressing the following issues.
Even if the questionnaire is intended to provide a subjective measure of exteroceptive awareness, beside the subjective measures of exteroceptive/interoceptive awareness, behavioural measures were used to test convergent and divergent validity. An explanation of the rationale for using these behavioural measures (to distinguish accuracy from sensibility and awareness level also in exteroception) would be useful. Moreover, predictions on results on convergent and divergent validity with the behavioural measures could help the readers.

In the introduction, the description of the scales used to test divergent validity seems too poor and not clear. “Subscales of the Body Consciousness questionnaire and the Self-Consciousness
questionnaire.” Please provide the references for the questionnaires. Throughout the manuscript, some acronyms already specified are not used (for example line 151 Body consciousness questionnaire must be BCQ, as specified in line 143; line 203 Self Consciousness Scale (SCS). Please use the acronyms if already specified or if you prefer to use the spelled name remove the acronyms). With respect to the Body Consciousness questionnaire, it would be important to provide the reader with an explanation of why private body consciousness scale and the public subscales were used as subjective measures of divergent and convergent validity.

A definition of exteroceptive awareness from the beginning of the introduction would be useful.

Line 168.169: A sample of 371 participants (XXX) volunteers. Should it be volunteer participants?

Line 180: “A sample of 184 participants (102 males, mean age = 25.8, standard deviation = 5.05)
recruited through the SCNLab voluntary database for another experiment participated in
the third study and completed the reduced 18-item EBA…”: are the authors referring to the third or to the second study?
“Sample of Study 3. A sample of 64 participants (26 males, mean age = 24.52, standard deviation = 4.98), recruited through the SCNLab voluntary database for another experiment, participated
in the second study aimed at characterizing the exteroceptive body awareness construct.”
Are the author referring to the third study?

For study 1 it has been specified that the study was carried online. I don’t find the information about where/how studies 2 and 3 were performed.

Line 216: there is a typo (.T )

Line 308: as already done for the interoceptive meta-awareness index (line 318), it would be useful to provide an explanation for interpreting the exteroceptive meta-awareness index.

Line 318. It emerges that participants also performed an evaluation through VAS about interoceptive accuracy. I did not find this information in the description of the task.

Results: 377-379: it would be useful to have a report of the main significant correlations also in the text.
It is not clear to me why Bayesian correlations were used only for the behavioural outcomes (and not for the other self-reported measures).

Do the authors also consider the correlations with the interoceptive meta-awareness index? These correlations have not been reported.

392-393
for the correlation between the interoceptive accuracy and the EBA questionnaire, a BF of 0.890 has been reported. Table 4 reported BF=0.378.

---

## Round 0.2 · accepted · Accept

The authors have addressed all of the reviewers' comments. One reviewer did not reply to the second round of revision, but he/she proposed a minor revision; after my own assessment of the paper, I can say I am happy with the current version. This manuscript is therefore ready for publication.

Reviewer 1 ·

Basic reporting

Reporting is fine

Experimental design

Fine.

Validity of the findings

Fine.

Additional comments

I want to thank the Authors for implementing the changes requested and for clarifying where that was necessary. I think it is good to go!
Please remember to make the OSF link public (not view-only) for the final version.